# A Comparative Evaluation of the Strain Transmitted through Prostheses on Implants with Two Different Macro-Structures and Connection during Insertion and Loading Phase: An In Vitro Study

**DOI:** 10.3390/ma15144954

**Published:** 2022-07-16

**Authors:** Riyam Kassem, Amar Samara, Ameer Biadsee, Shchada Masarwa, Tarek Mtanis, Zeev Ormianer

**Affiliations:** 1Department of Oral Rehabilitation, The Maurice and Gabriela Goldschleger School of Dental Medicine, Tel Aviv University, Ramat Aviv, Tel Aviv 6997801, Israel; riam_ka@hotmail.com (R.K.); ameerb@mail.tau.ac.il (A.B.); masarwas@mail.tau.ac.il (S.M.); tarekmtanis@mail.tau.ac.il (T.M.); 2Private Practice, Or Yehuda 4491500, Israel; ammar_smara@hotmail.com

**Keywords:** strain gauges, implant design, crown implant ratio, implant-abutment connection, bone

## Abstract

Background: The purpose of this study was to measure and compare the strain levels in the peri-implant bone as generated by the blade-like implant (BLI) and the screw-type implant (STI) with two different internal connections (hexagonal and conical) and with a 1:1 and 2:1 crown/implant (C/I) ratio. Methods: The implants (BLI and STI) were placed into sawbones according to the manufacturer’s protocol. Two strain gauges, horizontal and vertical to the implant axis, were placed around each implant on the bone surface 1 mm from the cervical part. Each implant was loaded by a material testing machine at a force of 100 N. Micro-strains (με) generated in the surrounding bone were measured by a strain gauge and recorded. Results: Recorded micro-strains were not significant in both the insertion and loading phases (*p* < 0.0625). The average recorded micro-strain values were lower in the horizontal dimension of STI with hexagonal connection when the C/I ratio was 2:1 compared with BLI, 210 με and 443 με, respectively. Conclusion: Within the limitations of this study, implant design, implant-abutment connection and C/I ratio did not influence strain values in bone and there is no statistically significant effect of these parameters on bone.

## 1. Introduction

Bone volume deficiency is a challenge in implant dentistry. Hence, a blade-like implant (BLI) may provide solutions in patients where there is severe osseous atrophy for which standard implant treatment cannot be conducted due to the presence of important anatomical landmarks [1,2,3,4]. The necessity for an alternative to traditional implants has led to the idea of endosseous BLI [1]. Subsequently, it was manufactured in a variety of shapes and prosthetic components to overcome these obstacles [5,6,7,8]. The clinical success or failure of a newly developed implant is related to the way that biomechanical features are transferred from the implant to the surrounding bone. 

The most common tool for evaluating biomechanical stresses is a strain gauge. Strain gauge analysis is a technique for micro-strain recording. After applying a force, the test specimen experiences strain; this strain is transferred directly to the strain gauge, which converts a resistance change to an electrical voltage [9].

Frost H.M. assumed that there is a regulatory mechanism that adapts bone mass according to the intensity of micro-strains (με) generated inside the osseous tissue at the bone–implant interface. For instance, if the mechanical stress does not reach the minimum functional value necessary for bone maintenance (below 50–200 με) or disuse of the bone leading to bone loss, 200 to 2500 με is equated with balanced bone remodeling, 2500 to 4000 με may trigger bone growth, and 3000 με or greater can theoretically lead to bone resorption [10].

One of the critical elements influencing the long-term uncompromised functioning of a dental implant is its design [11,12]. The shape of an implant can directly affect the distribution of stress and strain in both the implant and the surrounding bone [13,14]. A study showed that under axial load, especially in the low-density bone models, the maximum equivalent strain in cancellous bone was lower with the screw-type implant than with the cylinder-type implant [15]. Another study claimed that adequate bone stimulation via mechanical coupling may account for the larger bone response around the screw-type implant compared with the cylindrical implant [16]. However, the BLI effect on bone strain distribution has so far not been clearly confirmed by any well-designed and conducted clinical trials. 

An implant–abutment connection is one of the factors that significantly contributes to the load-bearing capacity of implants [17]. In vitro methods were used to compare differences in stress distribution patterns between implants with external or internal connections [18]. Studies have reported that the internal connection implants with a conical connection (Morse taper) exhibit a stable abutment connection, high resistance to axial loads and better stress distribution in the bone tissue when compared with an external hexagon connection [19,20,21,22]. However, this outcome remains controversial, as in some studies [23,24] conical connection did not reduce the micro-strain in the bone tissue around the implants when it was compared with the internal and external hexagonal connection. 

Excessive non-axis load that will increase the tendency to load the cervical supporting bone of the implant may be a result of an unfavorable crown/implant (C/I) ratio [25]. Increasing the crown length and degree of nonaxial load enlarges the risk of excessive occlusal overload due to an elongated moment arm [26]. Although it is commonly accepted that an unfavorable C/I ratio is potentially damaging, limited data are available on the effects of varying C/I ratios on stress transfer and distribution of the implants and their supporting structures. However, more recent studies revealed that the C/I ratio is not a reliable predictor of marginal bone loss or implant survival [27,28]. One study concluded that implant restorations may be successful even with C/I ratios between 1:2 and 1:3 and did not influence crestal bone loss [29]. In another study, a mean C/I ratio of 1.5:1 again did not affect crestal bone levels [30]. So, there are no guidelines to indicate a security threshold for an adequate C/I ratio.

The purpose of this study was to determine and compare the strain transmitted through prostheses of 8 mm and 16 mm lengths on implants of two different macrostructures (BLI and STI) and connections (hexagonal and conical) in the insertion and loading phases.

The null hypothesis is that differences in the strain distributions would be found among the two types of implants, connections, and C/I ratios. 

## 2. Materials and Methods

This study was carried out in the Department of Oral Rehabilitation in the Maurice and Gabriela Goldschleger School of Dental Medicine, Tel Aviv University. The insertion material used was made of a rigid polyurethane foam known as Sawbones (1522-105 block GP30# laminated on both sides w/3 mm) acquired from the company Sawbones (Vashon Island, Washington, WA, USA). Rigid foams were designed for use as an alternative biomechanical test material for cortical bone (Table 1). The density of the sawbones used was chosen by the present authors after carrying out an image analysis evaluation of the porosity of the cortical bone found in the different regions of the mandible.

Cubic samples of sawbones were designed as a cross-light shape with a dimension of 25 × 25 × 25 mm in order to bond the strain gauges as close as possible to the designed implant location. For the experimental tests of strain, two types of implants with 8 mm length were tested: the blade-like implant (BLI) (Startanius, Park Dental, Ardmore, OK USA) and the screw-type implant (STI) (Dynamix, Cortex Dental, Shlomi, Israel) with the conical and hexagonal abutment connections (Figure 1). In both implant types, the conical connection type exhibits 21.9° angulation and 2.8 mm diameter. The hexagonal connection exhibits an internal depth of 2.0 mm.

Eight groups were obtained:15 BLI with hexagonal connection.15 BLI with conical connection.15 STI with hexagonal connection.15 STI with conical connection.

An abutment of 8 mm (C/I 1:1) was connected to groups 1–4. 

An abutment of 16 mm (C/I 2:1) was connected to groups 5–8 (same implant types and configurations as groups 1–4) (Figure 2). The abutments were connected to the implants by a torque controller at 35 N/cm.

A hole was drilled at the sawbone sample to insert the dental implants. The preparation of the test specimens was made based on the surgical protocol and the drilling sequence recommended by the implant manufacturer. 

For the STI, the pilot drill was followed by 2.8 mm and 3.2 mm drills and for the BLI, a NRpieso (Park-dental, Ardmore, OK, USA) was employed to insert the implant. The implant insertion was carried out with a template mounting the implant at 15 degrees for non-axial loading. Vertical (grid 1) and horizontal (grid 2) strain gauges were placed at a 350 V nominal distance around each implant on the bone surface 1 mm from the cervical part (C2A-06-062LT-350; Vishay Measurements Group, Inc., Malvern, PA, USA). The measurement direction was parallel to the long axis of the implant. The strain gauges were connected to a strain indicator that took a measurement every 0.1 s (Figure 3). 

Strain was measured during the insertion phase for BLI and for STI. Then, each test specimen was mounted in an insertion loading machine (a special custom machine for Tel-Aviv University by J. MANAS LTD., Tel-Aviv, Israel) (Figure 4). Abutments of 8 mm and 16 mm were prepared to mimic 1:1 and 2:1 C/I ratios. The abutments were connected to the implants. Each sample was loaded with 100 N for 100 cycles [31,32].

### Statistical Analysis

A Mann–Whitney U test was performed to assess intergroup and intragroup comparisons among the mean:Intergroup comparison of insertion between STI and BLI.Intragroup comparison of hexagonal connection and conical connection values of the BLI and STI when the C/I ratios are 1:1 and 1:2.The same intragroup and intergroup comparisons were carried out for the maximum values of each measurement.

According to a sample size calculation: test significance = 0.05, power = 0.8, and effect size = 1.7.

The *p*-value was adjusted by using the Benjamini–Hochberg method for multiple comparisons.

## 3. Results

### 3.1. Insertion Phase

The implants in both groups were inserted according to the manufacturer’s protocol. During the test, a technical problem happened due to an electrical fault in a few strain gauges which caused the exclusion of some implants. The eliminated implants were not calculated in the statistical analysis.

The statistical analysis of the insertion phase of both implants indicated that implant geometry was not statistically significant in the horizontal dimension (*p* = 0.316) and in the vertical dimension (*p* = 0.0625) for the STI and the BLI (Table 2).

### 3.2. Loading Phase

Several strain–gauge disconnections occurred during the loading and reduced the number of the tested samples.

Within the BLI groups, no connection type—neither the hexagonal nor the conical—showed superiority over the other in the horizontal and the vertical moments when the C/I ratio was 1:1 or 2:1. Both were not statically significant (Table 3).

Similarly, within the STI groups, there were not statically significant in both types of connections in both dimensions and when C/I ratio was 1:1 or 2:1 (Table 4).

In the horizontal dimension, when the C/I ratios were 1:1 and 2:1, the STI groups showed the minimal average micro-strain in both types of connections, the hexagonal and the conical, with the superiority for the hexagonal connection that demonstrates the smallest average micro-strain values especially, when the C/I ratio was 2:1, but the results were not statistically significant. (*p* < 0.0625)

### 3.3. Maximum Loading Values

The statistical analysis for the maximum values shows that there is no statistical difference when the connection type was conical, or hexagonal or when the C/I ratio is 1:1 or 2:1 in both the vertical and the horizontal dimensions. It can be noticed that the STI groups demonstrate minimal values in the horizontal dimensions in both types of connections when the C/I ratio was 1:1 or 2:1 (Table 5 and Table 6).

Strain and high SD values were low in all the measurements.

The reason for this phenomenon is the load of 100 N which represents the clinical situation but exerts moderate µƐ values.

## 4. Discussion

One of the aims of this study was to analyze the influence of the insertion of different implant designs, BLI and STI, on the strain distribution in sawbones. The use of a polyurethane model is a cost-effective technique for evaluating the mechanical properties of dental implants. To eliminate the effect of any parameters on the implant insertion, the implants were inserted in homogenous material to ensure that the results are functions only of the implant design while overcoming the anatomical and ethical limits of ex vivo investigation. The results revealed that the strain distributions in the bone were noticeably not significantly affected by the implant design. Frost H.M. assumed that strain values above 200 με support bone remodeling and equilibrium [10]. All implant configurations demonstrate low με values in functional usage. The STI exerts more vertical strain in the insertion phase because of its tapered design. This is an important result because the implant itself is in direct contact with the bone and not separated from the bone tissue by the periodontal ligament as teeth are. Even the extended C/I ratio did not increase the με values. 

Brunski J.B. claimed that to achieve osteointegration, designers of implant systems must confront biomaterial and biomechanical subproblems, including in vivo forces on implants, load transmission to the interface, and interfacial tissue response [7]. 

Cochrane’s review of different types of dental implants did not demonstrate that any implant has superior long-term success over another [33]. Despite the fact that there were no significant micro-stain values between STI and BLI, it can be noticed that the STI showed to be the most favorable implant design involving stress and strain distribution to the bone in most simulated situations. A previous work concludes that screw-type implants could have a favorable response compared to cylinder implants, mainly in low-density bone [13]. Vandamme et al. concluded that sufficient bone stimulation via mechanical coupling may account for the large bone response around the screw-type implant compared with the cylindrical implant [14]. In another rabbit study, a more complete bone-to-implant contact around screw-shaped implants than around double cylinders and T-shaped implants was found [34]. However, the cylindrical implant type is not totally similar to the NRI, although the insertion method is the same. The implant diameter may have an influence on bone loss as demonstrated in several studies [35,36]. A wider implant diameter can distribute strain and occlusal forces better than a narrow-diameter implant. 

In the current study, the second aim was to investigate whether different implant–abutment geometries—conical or hexagonal—affect the strain distribution in bones for implants with different geometry. The results showed that there was no statistically significant difference between both connection types, but STI with a hexagonal connection demonstrates a minimal average micro-strain compared to a conical connection. It is widely debated that the implant–abutment connection design can induce different degrees of crestal bone remodeling. Caricasulo et al. reviewed the influence of the implant–abutment connection on peri-implant bone loss and found that internal connection, particularly the conical connection, exhibited a lower peri-implant loss in the short to medium term compared to an external connection [37]. An FEA study of an internal hexagonal connection implant and a conical connection implant under an applied force of 100 N found that the conical connection implant connected to a solid, internal, conical abutment and put lower stresses on the alveolar bone and prosthesis and greater stresses on the abutment relative to the internal hexed connection implant [38]. In other studies, external hexagon, internal hexagon and morse taper connection behavior under oblique and axial stresses of 100 N and 200 N including three inclinations (0°, 17°, and 30°) were compared; the morse taper implants showed biomechanical superiority more than any other connections, especially during oblique loading [19,20]. This current study does not support these findings, perhaps due to the non-axial loading that was employed and the macro-structures of different implant types that were used. 

Conversely, another study compared internal cone, internal and external hexagonal connections for strain/stress distribution around implants, and compared the effects of implant–abutment connections and implant fixture alignment. Morse tapers and internal hexagons did not reduce strain around the implants; no statistical significance in the placement configuration was observed and it has been suggested that the bone loss is a consequence of the lack of mechanical coupling between the machined coronal region of the implant and the bone, which prevents the effective transfer of occlusal forces from the implant to the cortical bone [23]. Support for this statement can be found in another RCT study, which concludes that there are no statistically or clinically significant differences observed between the two types of internal connections 1 year after loading [24]. Esposito et al. compared screw-shaped implants with external or internal connections and revealed that there are no significant differences even 1 and 5 years after loading [39,40]. Both implants gradually lost an average of 1.13 mm peri-implant bone for external connection and an average of 1.21 mm bone for internal connection. The non-significant differences can be explained by the internal hexagonal connection reducing the probability of micro-movement during loading, similar to the conical design.

The present study was conducted to evaluate the strain distribution when C/I ratios were 1:1 and 2:1 for different implant–abutment connection geometries (hexagonal, conical) for both implants—the BLI and STI. The minimal average strain developed around the STI in the horizontal dimension when the C/I ratios were 1:1 and 2:1 for micro-strains in both types of connections. The hexagonal connection demonstrates the smallest average micro-strain values, especially when the C/I was 2:1. These findings support other studies which demonstrated that an increased C/I ratio may not influence marginal bone loss. One study has shown that the restoration may be successful even with a C/I ratio between 2:1 and 3:1 (29). Another study investigated 326 implants with a mean C/I ratio of 1.6, and observed that an excessive C/I ratio had no negative effect on the peri-implant bone loss but caused more significant prosthetic complications such as screw loosening and porcelain fractures [41].

The limitations of this study are the sawbone model and the relatively low strain measurements. Although the sawbone model is an acceptable model in many studies, it does not always reflect the native bone itself. The low strain measurement can be explained by the 100 N load that was applied to mimic the jaw chewing force. The combination of both may cause the low strain values.

## 5. Conclusions

With the insertion of different macro-structure designs, BLI and STI implants did not exhibit significantly different strain distributions in the sawbone model. The implant– abutment connection type as well as the C/I ratio seems to have no effect on strain distribution in the sawbone model. Further controlled studies with an increased C/I ratio on the impact of horizontal movements on implant–abatment connections as observed in removable restorations are needed.

## Figures and Tables

**Figure 1 materials-15-04954-f001:**
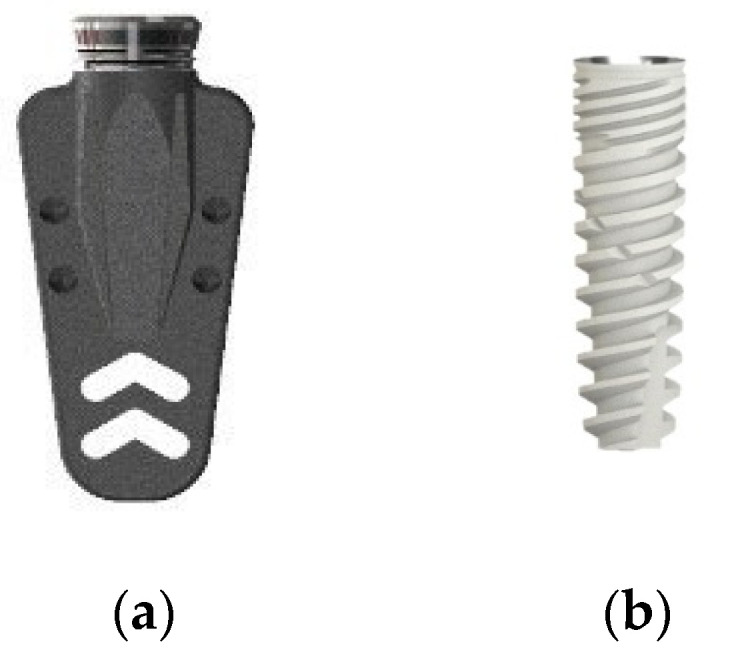
Implant system used in the experimental test (**a**) BlI and (**b**) STI.

**Figure 2 materials-15-04954-f002:**
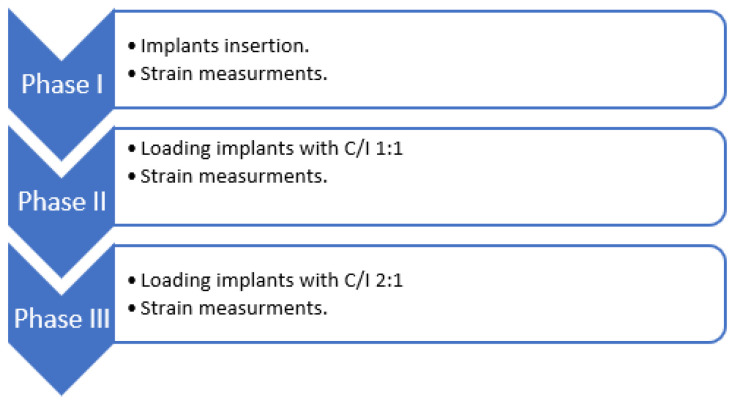
A diagram of the experiment design.

**Figure 3 materials-15-04954-f003:**
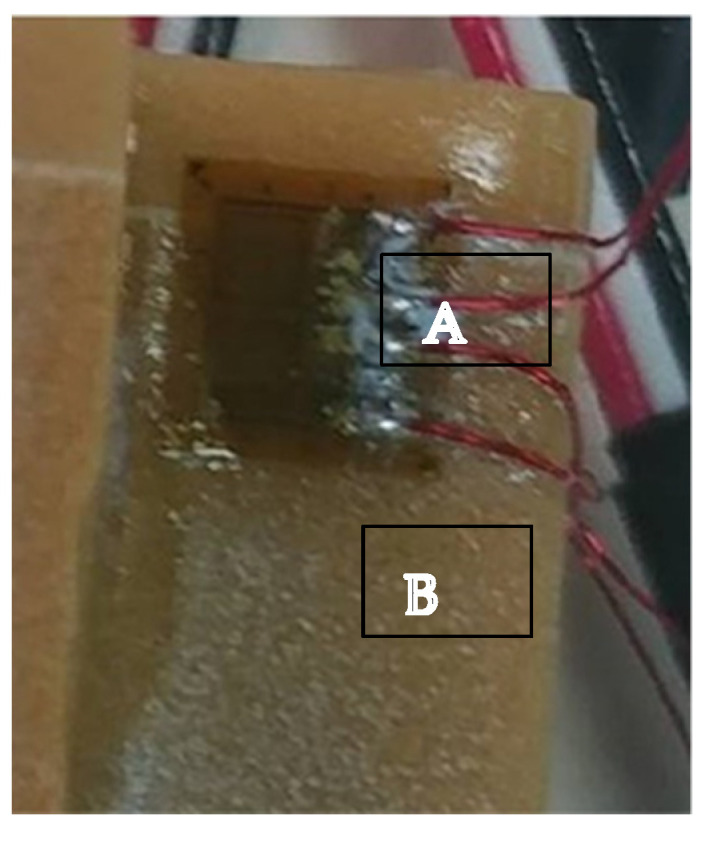
The strain gauges (**A**) connected directly to the sawbone (**B**) surface in the cervical part of the implant.

**Figure 4 materials-15-04954-f004:**
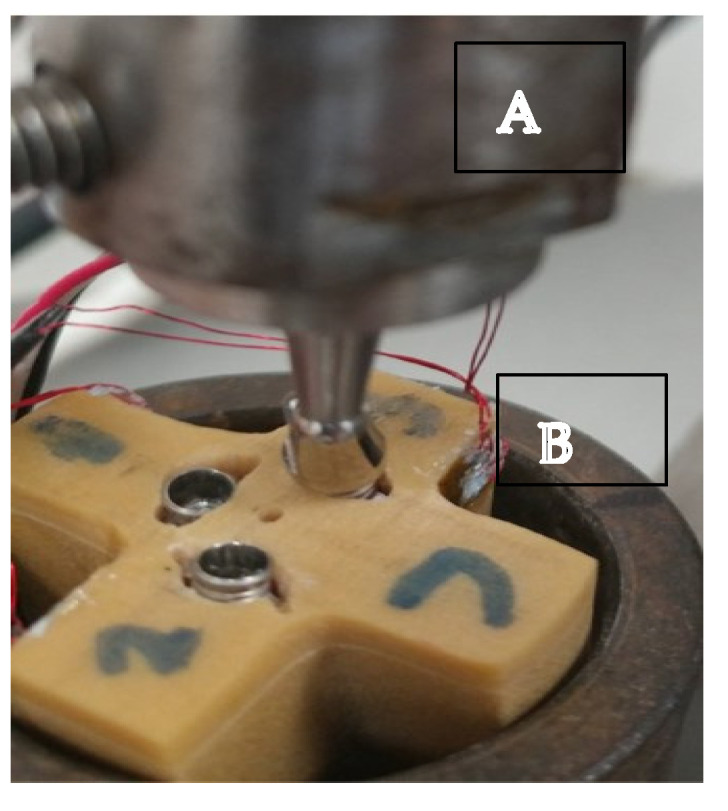
The insertion loading machine (**A**) creates non-axial pressure on the abutment (**B**) connected to the implant.

**Table 1 materials-15-04954-t001:** Simulated cortical bone strength (data taken from Sawbones catalog www.sawbones.com/catalog/biomechanical/blocks-and-sheets (accessed on 1 July 2022)).

Density	Longitudinal Tensile		Transverse Tensile		Compressive	
	Strength	Modulus	Strength	Modulus	Strength	Modulus
(g/cc)	(MPa)	(GPa)	(MPa)	(GPa)	(MPa)	(GPa)
1.64	106	16	93	10	157	16.7

**Table 2 materials-15-04954-t002:** Comparison of strain in the insertion phase between STI and BLI. Vertical με is lower in both implants.

	BLI (N = 29)	STI (N = 29)	PV
**Horizontal**			<0.316
Mean (με) ± SD	681.16 ± 920.74	580.95 ± 402.46	
Min.–Max.	2.97–3840.73	1.74–1674	
Median	29.35	564.31	
Percentiles 25/75	−201.64/320.15	245.45/782.42	
**Vertical**			<0.0625
Mean (με) ± SD	129.78 ± 179.09	224.68 ± 159.15	
Min.–Max.	1.46–697.17	5.92–611.98	
Median	−4.26	−205.18	
Percentiles 25/75	−68.12/62.63	−311.05/−90.45	

V/H—vertical and horizontal strain–gauge.

**Table 3 materials-15-04954-t003:** Intragroup comparison of hexagonal connection and conical connection values of the BLI when C/I ratio is 1:1 and 2:1. με are related in both connections and C/I ratio at the same implant configuration.

	Conical (N = 10)	Hexagonal (N = 14)	PV
**Vertical**			<0.259
C/I 1:1			
Mean (με) ± SD	66.38 ± 53.24	42.098 ± 46.78	
Min.–Max.	11.9–154.07	2.99–172.66	
Median	−38.7	−19.42	
Percentiles 25/75	−113.38/−9.93	−36.5/10.07	
**Horizontal**			<0.931
C/I 1:1			
Mean (με) ± SD	111.96 ± 136.48	85.71 ± 65.13	
Min.–Max.	4.71–400.7	2.044–177.84	
Median	27.98	49.18	
Percentiles 25/75	−42.51/74.15	−18.55/150.82	
**Vertical**			<0.064
C/I 2:1			
Mean (με) ± SD	88.59 ± 45.92	56.47 ± 39.19	
Min.–Max.	30.83–151.39	3.13–129.7	
Median	−67.91	−27.79	
Percentiles 25/75	−136.47/34.74	−79.53/19.36	
**Horizontal**			<0.709
C/I 2:1			
Mean (με) ± SD	169.60 ± 209.13	168.76 ± 202.63	
Min.–Max.	42.34–625.28	4.28–662.22	
Median	52.27	95.63	
Percentiles 25/75	−56.64/96.25	36.03/208.5	

V/H—vertical and horizontal strain–gauge.

**Table 4 materials-15-04954-t004:** Intragroup comparison of hexagonal and conical connection values of the STI when C/I tatio is 1:1 and 2:1. με are related in both connections and C/I ratio at the same implant configuration.

	Conical (N = 15)	Hexagonal (N = 13)	PV
**Vertical**			<0.821
C/I 1:1			
Mean (με) ± SD	60.16 ± 39.61	52.99 ± 37.60	
Min.–Max.	2.43–129.29	5.89–102.65	
Median	−69.24	−41.147	
Percentiles 25/75	−88.3/−15.2	−87.28/9.15	
**Horizontal**			<0.294
C/I 1:1			
Mean (με) ± SD	48.8 ± 53.78	60.30 ± 47.42	
Min.–Max.	0.88–217.22	10.18–171.44	
Median	0.88	−21.36	
Percentiles 25/75	−38.44/50.04	−77.66/22.96	
**Vertical**			<0.786
C/I 2:1			
Mean (με) ± SD	80.58 ± 74.04	60.40 ± 50.61	
Min.–Max.	3.31–243.46	3.47–194.05	
Median	−51.26	−46.8	
Percentiles 25/75	−108.79/−16.39	−79.19/18.09	
**Horizontal**			<0.235
C/I 2:1			
Mean (με) ± SD	43.7 ± 30.71	35.27 ± 36.99	
Min.–Max.	8.98–110.81	0.36–119.63	
Median	8.98	7.87	
Percentiles 25/75	−46.21/36.19	−36.67/21.30	

V/H—vertical and horizontal strain–gauge.

**Table 5 materials-15-04954-t005:** Intragroup comparison of hexagonal and conical connection maximum values of the BLI when C/I ratio is 1:1 and 2:1.

	Conical (N = 10)	Hexagonal (N = 14)	PV
**Vertical**			<0.192
C/I 1:1			
Mean (με) ± SD	307 ± 187.82	219.57 ± 162.01	
Min.–Max.	78–668	58–548	
Median	273	143	
Percentiles 25/75	157/487	116.50/396	
**Horizontal**			<0.886
C/I 1:1			
Mean (με) ± SD	325.9 ± 221.52	358.21 ± 270.14	
Min.–Max.	127–854	76–885	
Median	271	303	
Percentiles 25/75	170/419.75	133/524.75	
**Vertical**			<0.403
C/I 2:1			
Mean (με) ± SD	385.4 ± 201.33	318.29 ± 159.36	
Min.–Max.	129–737	111–590	
Median	378.50	281	
Percentiles 25/75	196.50/534.25	173.75/468.25	
**Horizontal**			<0.841
C/I 2:1			
Mean (με) ± SD	443.5 ± 235.24	516.93 ± 484.09	
Min.–Max.	176–917	72–1778	
Median	376.50	349	
Percentiles 25/75	271.25/618.25	190.50/742	

V/H—vertical and horizontal strain–gauge.

**Table 6 materials-15-04954-t006:** Intragroup comparison of hexagonal and conical connection maximum values of the STI when C/I ratio is 1:1 and 2:1.

	Conical (N = 15)	Hexagonal (N = 13)	PV
**Vertical**			<0.928
C/I 1:1			
Mean (με) ± SD	323.67 ± 154.65	320.92 ± 147.50	
Min.–Max.	105–605	86–526	
Median	273	317	
Percentiles 25/75	190/445	197/446.50	
**Horizontal**			<0.363
C/I 1:1			
Mean (με) ± SD	187.53 ± 77.26	263.38 ± 178.95	
Min.–Max.	90–335	82–679	
Median	147	181	
Percentiles 25/75	131/251	124/398	
**Vertical**			<0.821
C/I 2:1			
Mean (με) ± SD	413.67 ± 301.38	351.92 ± 160.31	
Min.–Max.	124–1001	130–723	
Median	286	326	
Percentiles 25/75	173/559	229.50/423	
**Horizontal**			<0.339
C/I 2:1			
Mean (με) ± SD	220.33 ± 131.8	210.08 ± 169.081	
Min.–Max.	83–534	59–601	
Median	188	137	
Percentiles 25/75	125/252	100/310.50	

V/H—vertical and horizontal strain–gauge.

## Data Availability

Data is contained within the article.

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
