# Peer review of "A Comparative Evaluation of the Strain Transmitted through Prostheses on Implants with Two Different Macro-Structures and Connection during Insertion and Loading Phase: An In Vitro Study"

_materials, 2022, doi:10.3390/ma15144954_

Round 1

Reviewer 1 Report

This in-vitro study has explored effects of implant macro-structure, implant-abutment connection, and C/I ratio on bone strain distribution in the insertion and loading phase. The authors found there was no statistically significant influence for these parameters on bone tissue surrounding implants. And here are some suggestions about this manuscript:

  1. The bone strain surrounding implants was related to MBL and several compilations of implant-supported prosthesis and authors should review some literature and mention this point in the introduction
  2. materials and methods
  • According to the different implant-abutment connection of two implants (BLI, STI), part, it is necessary to state the exact taper, respectively.
  • As for the C/I ratio, it should be 2:1 when the abutment was 16 mm, not the 1:2. Also, the subgroup of C/I ratio should be separated further.
  • The torque of abutment connecting on implants should be stated.
  • According to the statistical analysis, the method about normal distribution analysis need adding, in the Mann-Whitney U Test, medians and quartiles should be presented.
  1. results
  • As for the strain-gauge disconnections occurred during loading, it should be described in detail.
  • Table 5 was the integration of table 3-4 and it seems a little redundant; why put the different connections on the same line in table 5.
  • In table 6-7, it did not present the difference between two implants and authors might consider another format, such as bar graph, scatter plot to point these results.
  1. discussion
  • In the insertion phase, STI implant exhibited higher vertical strain, and this should be discussed in this part.
  • Why the abutment connections had different influence on vertical bone strain in two implants, and authors should not ignore specific parameters of implant shape to analyze this point.

Author Response

Thank you for your comments.  the answers are in the attached file.

Reviewer 2 Report

The paper entitled, The effects on bone strain distribution on implant microstructure and prosthetic configurations.  The following aspects need to be addressed

Title

  1. Rewrite the title to reflect the content of the paper

Abstract

  1. Add introduction to show the important of the study and problems of the existing study.

Introduction

  1. Line 74, and Figure C/I ratio so that the reader can see it clearly

Material and Methods

  1. Line 99, show figure of experimental setup
  2. Line 107, provide citation for Sawbone catalogue
  3. Figure 1, line 114: Show size, dimensions, and scale for both BLI and STI.
  4. Line 115- line 124. Rewrite the sentences.  It is quite confusing.
  5. Line 126- Provide citation of surgical protocol
  6. Line 127 – Provide citation drilling sequence by manufacturer
  7. Line 135, Figure 2 – Label important item
  8. Line 144, Figure 3 – Label important item
  9. Line 147, Statistical analysis – cite references for Mann-Whitney U Test
  10. Line 154, spelling error: According
  11. Line 156, cite references for Benjamini-Hochberg method
  12. Line 160, spelling error: Manufacture
  13. Line 166, spelling error: Vertical…

Discussion

  1. Line 5, spelling error: assumed
  2. Line 8, spelling error: separated
  3. The result is not discussed in depth. There is no FE work been done and compared.
  4. There is no validation of the work done. It is expected to the author to show how they validate the work. 

References

21. References used are not current.

Author Response

(The authors gave the same response as above.)

Reviewer 3 Report

Thank you for the opportunity to review paper “Implant macro-structure and prosthetic configurations and its effect on bone strain distribution”

In my opinion before the publishing the paper needs extensive correction.

The title is confusing it should be emphasize that this is in vitro study and not performed on bone but on a artificial polyurethane blocks (which are more uniform and represent more of the cortical than on the spongiouse  bone).

 36-38-  The references are interesting but they don’t reflect the statement in the text for example in the article  “Immediately Loaded Blade Implant Retrieved From a Man After a 20-year Loading Period: A Histologic and Histomorphometric Case Report”  there are no information that it was severe osseous atrophy – some references should be removed and other choose more precisely.

39-40-what dose it means that there was no indication for use ? please visit http://www.linkowlibrary.org/

43- to many references as they refer to the same design

44- What does it mean Frost ? Should be HM. Frost

52-53- it should be changed as the design is not the main factor for implant survival; those references are old and should be more updated.

In the introduction when they describe factor/ elements influencing the long-term uncompromised  functioning of a dental implant and MBL they neglect the tilted implants used in many methods and systems like all-on -four or tubero-pterygoid implants .That is why the introduction should be rewritten and should focus more on bone physiology and its reaction on different shapes and surfaces of the implants. I would recommend the authors to read papers of Skalak , Brunski or Weiss.

The aims ,especially the first one “To investigate whether different implant geometry induced different  strain values when it is inserted into bone is extremely confusing. There are many types of bone , with different architecture , function and mineralization. It needs to be precisely explained which type of bone and why this one. The authors did not explain why exactly those two types were chosen , why this length and diameter and different hex size.

Where can I find information that BLI implant is grade V titanium ?The BLI implants are “piezo placed “ so the bed preparation is completely different.

There is no information about insertion torque or primary stability of the implants.

The literature in the references is quite old and should be up-dated with more contemporary research. In my opinion articles in this shape have a very low scientific value.

The design and presentation is poor thus it does not justify publication in high IF journals.

Author Response

Thank you for your comments. the answers are in the attached file 

Round 2

Reviewer 1 Report

The present revised manuscript titled “A comparative evaluation of the strain transmitted 2 through prostheses on implants with two different 3 macro-structures and connection during insertion and loading phase: An in-vitro study” reported the effects of implant macro-structure, implant-abutment connection, and C/I ratio on bone strain distribution in the insertion and loading phase. The manuscript was well revised, and this topic was important for the clinical guidance, but there were some minor limitations about this paper.

1.      According to the statistical analysis, the method about normal distribution analysis need adding, in the Mann-Whitney U Test, medians and quartiles should be presented. The author should not ignore this point.

2.      Some spelling errors should be corrected, such as the conclusion in the part of abstract (Implant should be implant). Also, descriptive errors should be avoided in the revised manuscript, such as the C/I ratio.

3.      The table legends should be corrected since table 5 was deleted.

Author Response

According to the statistical analysis, the method about normal distribution analysis need adding, in the Mann-Whitney U Test, medians and quartiles should be presented. The author should not ignore this point.

Answer: medians and quartiles were added in all tables 

  1. Some spelling errors should be corrected, such as the conclusion in the part of abstract (Implant should be implant). Also, descriptive errors should be avoided in the revised manuscript, such as the C/I ratio.

Answer: done 

  1. The table legends should be corrected since table 5 was deleted.

Answer: done

Thank you